# Quantitative real-time in-cell imaging reveals heterogeneous clusters of proteins prior to condensation

Chenyang Lan[1,2,3], Juhyeong Kim[1], Svenja Ulferts [4], Fernando Aprile-Garcia [5], Sophie Weyrauch[1,6,7], Abhinaya Anandamurugan[1], Robert Grosse [4], Ritwick Sawarkar [8], Aleks Reinhardt [9] ✉ & Thorsten Hugel [1,2] ✉

Our current understanding of biomolecular condensate formation is largely based on observing the final near-equilibrium condensate state. Despite expectations from classical nucleation theory, pre-critical protein clusters were recently shown to form under subsaturation conditions in vitro; if similar long-lived clusters comprising more than a few molecules are also present in cells, our understanding of the physical basis of biological phase separation may fundamentally change. Here, we combine fluorescence microscopy with photobleaching analysis to quantify the formation of clusters of NELF proteins in living, stressed cells. We categorise small and large clusters based on their dynamics and their response to p38 kinase inhibition. We find a broad distribution of pre-condensate cluster sizes and show that NELF protein cluster formation can be explained as non-classical nucleation with a surprisingly flat free-energy landscape for a wide range of sizes and an inhibition of condensation in unstressed cells.

Key steps of gene expression, including transcription[1–3], translation, as well as signalling[4–6] and metabolism[7,8], are regulated by membraneless assemblies of relevant macromolecules, namely proteins and nucleic acids. Such membraneless assemblies, often called condensates, have the advantage of rapid material exchange with their surroundings while keeping the macromolecules in spatial proximity[9–11]. They are thought to form by phase separation, with macromolecular concentrations higher within them than outside[10,12,13]. A key question in the field is how proteins form condensates at a molecular level[14–16]. Intriguingly, misregulated condensation has been shown to be causally linked with human pathologies[12,17–19]. A molecular understanding of the process of condensation is thus important from both a fundamental[12,20–25] and a biomedical perspective[26,27].

The question of how protein condensation occurs at the molecular level can be broken down into two related issues. First, it is not clear how proteins behave prior to the formation of condensates or at conditions at which condensates do not form. Second, it would be useful to know what properties of proteins change during condensate formation. In the traditional homogeneous nucleation picture of condensate formation, there is a competition between a favourable bulk term and a disfavourable surface term associated with the formation of an interface between the cluster and its surroundings[28]. The free energy of a cluster rapidly increases until a critical cluster size is reached, and then decreases as the bulk term begins to dominate[28]; clusters are therefore most likely either post-critical or very small[14]. In subsaturated conditions, the bulk term is itself disfavourable and only

[1]Institute of Physical Chemistry, University of Freiburg, Freiburg, Germany. [2]BIOSS and CIBSS Signalling Research Centres, University of Freiburg, Freiburg, Germany. [3]PicoQuant GmbH, Rudower Chaussee 29, 12489 Berlin, Germany. [4]Institute of Experimental and Clinical Pharmacology and Toxicology, University of Freiburg, Freiburg, Germany. [5]Max Planck Institute of Immunobiology and Epigenetics, Breisgau, Germany. [6]Spemann Graduate School of Biology and Medicine (SGBM), University of Freiburg, Freiburg, Germany. [7]Faculty of Chemistry and Pharmacology, University of Freiburg, Freiburg, Germany. [8]Medical Research Council (MRC), University of Cambridge, Cambridge CB2 1QR, United Kingdom. [9]Yusuf Hamied Department of Chemistry, University of Cambridge, Cambridge CB2 1EW, United Kingdom. ✉e-mail: ar732@cam.ac.uk; th@pc.uni-freiburg.de

small fluctuations in cluster size around the homogeneous monomeric phase occur. Within the complex cellular milieu of diverse macromolecules, there might be deviations from this ideal behaviour because of molecular crowding, the existence of inflexible polymers such as the cytoskeleton and a plethora of small-molecule metabolites. In addition, recent in vitro studies have revealed heterogeneous distributions of clusters in subsaturated solutions[29]. Moreover, the non-equilibrium environment of the cell allows for the circumvention of thermodynamic constraints and the emergence of new features, such as dynamic cluster (or droplet) localisation, which can arise in active systems[30]. Therefore, investigation of cluster formation within living cells is crucial.

Despite some recent successes[1,2,9,31–36], the quantification of the dynamics of cluster formation in living cells is still difficult because the study of proteins prior to condensate formation inside living cells is limited by several technical impediments. First, the signal-to-noise ratio in fluorescence measurements in living cells is low, and counting the number of fluorescent proteins is therefore usually done in fixed cells. Second, the density of proteins in clusters is high, which impedes the separation and counting of single proteins even with super-resolution imaging[1]. If sub-stoichiometric labelling is used, the bandwidth is limited, and either the process of cluster formation cannot be observed or the condensates cannot be quantified. Third, proteins within cells are mobile and dynamic, necessitating a high time resolution that is currently difficult to achieve with commercial setups at a low signal-to-noise ratio. In addition to these technical reasons, it is difficult to capture proteins in their non-equilibrium pre-condensate state. While in vitro studies can rely on titrating concentrations of proteins below the saturation threshold to observe such states, it is not straightforward to control protein levels inside cells. Given these limitations, most studies thus far have largely focused on the late (equilibrium) states of condensates even when small transient clusters or oligomers prior to condensate formation were detected[1,37], leaving a gap in our understanding of the pre-condensate behaviour of proteins.

These limitations and the changing nature of the composition of the cell make it even more difficult to determine precisely where the phase boundary is. On the other hand, the passive thermodynamic classification of sub- and super-saturated solutions is perhaps less important here, as nucleation only occurs under super-saturated conditions. For describing phase separation in living cells, it is more important to determine the free-energy landscape and to be able to separate condensate and pre-condensate states in time, i.e. to have a controlled, signalling-induced transition of proteins to condensates. An example of such a process is the stress-induced condensation of NELF (negative elongation factor), a nuclear transcriptional regulator. NELF has been closely linked to stress-induced transcriptional attenuation (SITA); moreover, p38 kinase signalling has been connected to gene downregulation[38]. Simple and controlled stressors such as $As_2O_3$ cause NELF to form condensates, leading to a global down-regulation of transcription[39]. The NELF complex comprises four subunits, with NELFA possessing an intrinsically disordered region and NELFE a receptor-binding domain[38,40]. Expression of NELFA-GFP enables NELF condensation to be observed upon arsenic stress in real time in living cells.

In this study, we combine super-resolution imaging and single-molecule microscopy in fixed and living cells to quantify the behaviour of NELF in cells both before and during condensation. We also investigate the effect of a p38 kinase inhibitor[38] on this process.

## Results

### Dynamic clusters can be tracked and analysed in living cells

Tracking cluster growth at near single-molecule sensitivity and a high time resolution requires the overall concentration of the tracked molecules to be low. To this end, we used a tetracycline-inducible system in HeLa cells to achieve low expression levels of NELFA-GFP[39].

We identified conditions under which NELFA-GFP is expressed to levels of ~25% of the endogenous NELFA in HeLa cells [Supplementary Note 1]. The condensation of NELFA was triggered by toxic stress (100 μM $As_2O_3$), which has been shown to result in similar condensation as heat stress[39]. We ensured that this treatment did not compromise cell viability [Supplementary Note 2].

The effect of NELFA-GFP concentration on condensation is shown in Fig. 1. At low expression levels, upon exposure to arsenic, several small clusters of NELFA-GFP, but only a single large condensate, were visible, while many large condensates occurred at high expression levels. Such a dependence on concentration is expected for nucleation and, in turn, for condensate formation. In order to be able to image and track single clusters, we select cells at low expression, since it becomes progressively more difficult to distinguish unambiguously between condensates and the dilute phase with increasing expression conditions; indeed, at high expression conditions (Fig. 1c), the measured cluster area depends on the imaging contrast. Most importantly, because we work at low expression, we show below that we are able to deduce two clear criteria to distinguish small clusters from condensates; namely, one entailing cluster-size dynamics and another the response to a kinase inhibitor.

Real-time tracking of cluster dynamics is not possible over extended times with Airyscan microscopes or other super-resolution imaging methods because of limitations in image acquisition time, poor signal-to-noise ratios, or strong photobleaching. To image and track NELFA-GFP in living cells, we therefore used a highly inclined and laminated optical sheet (HILO) microscope[41] [Fig. 1e; see also Supplementary Note 3]. We used camera exposure times of 70 ms, frame rates of $0.1 \, s^{-1}$ and low laser power to obtain sharp images and to avoid photobleaching. However, a low laser power also results in a reduced signal-to-noise ratio, rendering conventional threshold-based image analysis ineffective. We instead used a machine-learning algorithm to segment NELFA-GFP from cellular and non-cellular backgrounds, coupled with a single-particle tracking algorithm to track individual NELFA-GFP clusters during condensate formation [Supplementary Note 4].

Using this setup, we first imaged an unstressed living cell at intervals of 10 s. We observed the diffusion of individual NELFA-GFP spots until photobleaching occurred. Figure 1f shows four images of a cell without arsenic exposure alongside the corresponding cluster tracking analysis. Full trajectories for this and two additional cells are provided in Supplementary Movie 2, and the tracking analysis for the presented cell is provided in Supplementary Movie 3.

Next, we added 100 μM $As_2O_3$ to stress the cells and trigger the condensate formation of NELFA-GFP. In Fig. 1g, we show representative images along a trajectory for one cell as a function of time following exposure to $As_2O_3$. Many small NELFA-GFP clusters could be observed; these not only move in space, but also dynamically grow and shrink, which is quantified below. Full trajectories for this and eight additional cells are shown in Supplementary Movie 4, and the tracking analysis for this cell is provided in Supplementary Movie 5. In all cells in which NELFA condensates formed, we found that NELFA-GFP clusters continually grow and shrink until they reach a certain size (see Supplementary Note 5 for data on all cells). However, it appears that once a cluster reaches this size, it continues to grow into a larger condensate, and such dynamic behaviour can therefore serve as an initial distinction between small (pre-condensate) clusters and large clusters (condensates). At higher expression conditions, several clusters can reach the critical size (see Fig. 1a, c), which is an expected nucleation-like behaviour and is further investigated below.

### Fixed cells provide a calibration for living-cell data

Our microscope is capable of observing single GFPs; however, even at the lowest expression conditions studied, where only single visible condensates ultimately formed, the density within clusters soon

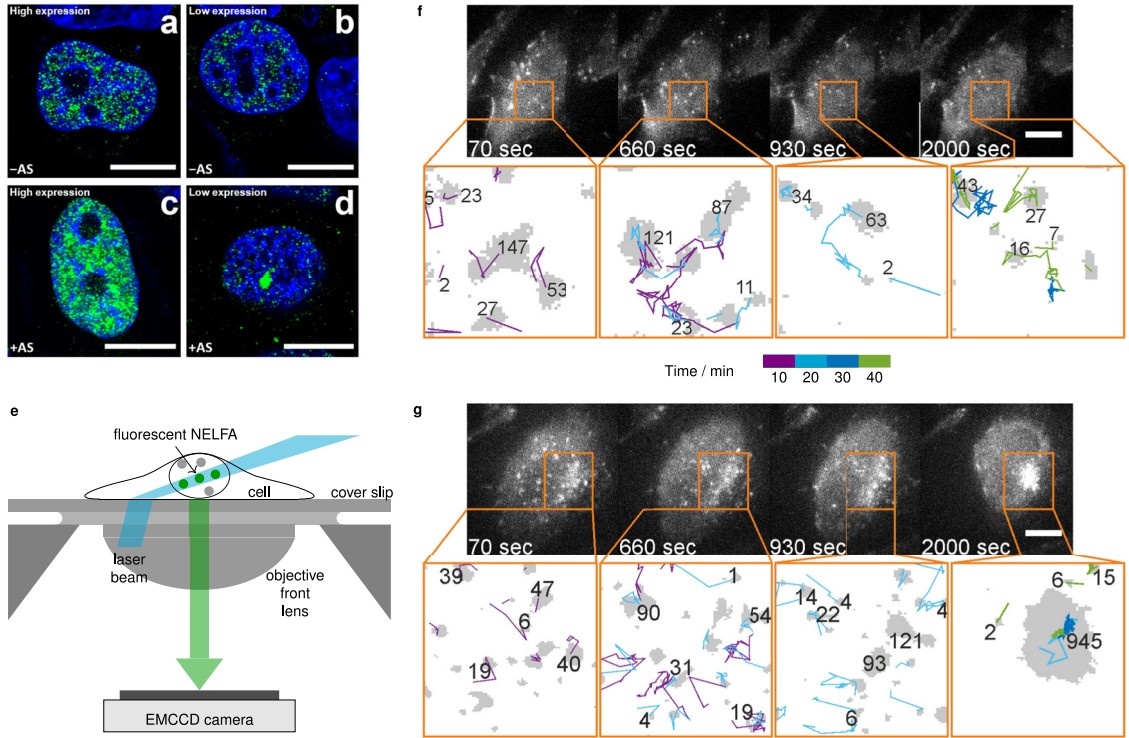

**Fig. 1 | Time-resolved imaging of clusters and condensates in cells. a–d** High-resolution Airyscan 3D-imaging of HeLa cells with NELFA-GFP at standard/high (**a**, **c**) and low (**b**, **d**) expression levels in the absence of arsenic stress (**a**, **b**) and following a 1-h exposure to As$_2$O$_3$ (**c**, **d**). NELFA-GFP is shown in green and nuclear DNA is stained with DAPI (blue). Scale bars 10 μm. See Supplementary Movie 1 for a *z*-scan view of representative cells with NELFA-GFP at high and low expressions. The red colour in the movie is a stain for the nuclear lamina with AF647. **e** Schematic illustration of the living-cell experiment. **f** Imaging of an unstressed HeLa cell with diffusing NELFA-GFP. Bright spots correspond to diffusing NELFA-GFP molecules imaged at different time points (as indicated). Camera exposure time 70 ms. Scale bar 10 μm. Outputs from image analysis are shown below microscopy images. NELFA-GFP clusters are shown in grey, with trailing lines identifying the track of each cluster. The number of NELFA-GFP within each cluster is shown alongside. The pixel size is 160 nm × 160 nm. Three cells; 679 tracks. **g** Analogous results for a HeLa cell with NELFA-GFP upon arsenic stress. The time after arsenic exposure is indicated. After about 930 s, one cluster grows irreversibly into a condensate. Camera exposure time 70 ms. Scale bar 10 μm. Nine cells; 3631 tracks. See Supplementary Movies 2–5 for time-resolved microscopy and image analysis of these and other cells.

became too high to separate and count single GFPs. We therefore combined our living-cell tracking experiments with photobleaching-step counting[42] in different fixed cells under identical expression and stress conditions, which enabled us to quantify the number of NELFA-GFP molecules in dense regions with near single-molecule accuracy. We added this quantification to the living-cell movies and images shown in Fig. 1f, g.

To obtain such data with near single-molecule sensitivity, we fixed HeLa cells at up to ten different times following exposure to As$_2$O$_3$ and counted photobleaching steps for all clusters. This cannot be readily done in living cells, as GFPs can only be bleached once at a defined time point. Counting photobleaching steps has the advantage of counting the local number of proteins with high accuracy; by contrast, in intensity-based measurements, brightness variations in cells affect the result. Figure 2a shows two examples of how single GFPs are counted in fixed cells over time. Since every photobleaching step corresponds to precisely one NELFA-GFP molecule, we can convert NELFA-GFP areas from living-cell imaging into the numbers of molecules in such an area, and in turn, obtain the number of molecules in each cluster. We double-checked this conversion with an intensity-based analysis (see Supplementary Note 6).

In Fig. 2c, we show that the areas of clusters at different time points for the nine living cells (coloured dots) agree well with the average data from the 35 fixed cells (black line), which indicates that the data from fixed and living cells are consistent. Such consistency is especially striking considering that the latter should be a lower limit on the condensate size, as the 35 fixed cells were selected randomly, and a

condensate would not have formed in all of them within 60 min. Finally, Supplementary Note 6 [Supplementary Fig. 6c] shows that the average density of NELFA-GFP in the dense phase (i.e. in clusters and condensates) is almost unchanging at ~29 μm$^{-2}$.

Living-cell experiments allow us to define a critical area necessary for condensate growth. We can determine for each cell the largest size of clusters that do not form a condensate, i.e. the size up to which cluster sizes fluctuate dynamically. We show examples of cluster size fluctuations in Fig. 2b; all clusters except the ones shown in violet and red eventually shrink. [Cluster-size data for all cells are shown in Supplementary Note 7.] For the nine cells, we have the following maximum size for dynamic clusters (from cell 1 to 9 in μm$^2$): 13, 17, 14, 12, 10, 19, 17, 13, (28), averaging (14 ± 3) μm$^2$. We consider the last value from cell 9 in brackets an outlier, because fluctuations occur only 40 min after arsenic exposure when the cell starts moving considerably. 14 μm$^2$ translates to ~400 NELFA-GFP (14 μm$^2$ × 29 molecules μm$^{-2}$). As only about every fifth molecule was GFP-labelled [Supplementary Fig. 1], the mean critical nucleus size under these conditions was thus ~2000 NELFA molecules. To validate our results, we determined the total number of NELFA molecules in the nucleus. To this end, we first determined the number of NELFA-GFP in the focal plane, which is roughly 1000 for the cell depicted in Fig. 1g and up to 5000 for the largest condensate observed (cell 1 in Supplementary Movie 4). The focal plane is ~1 μm deep; for a nucleus of ~7 μm in diameter, we therefore only detect about 1/5 of the molecules. Moreover, since only every fifth molecule was labelled, we multiply the number of molecules in the focal plane by 25 in total, resulting in

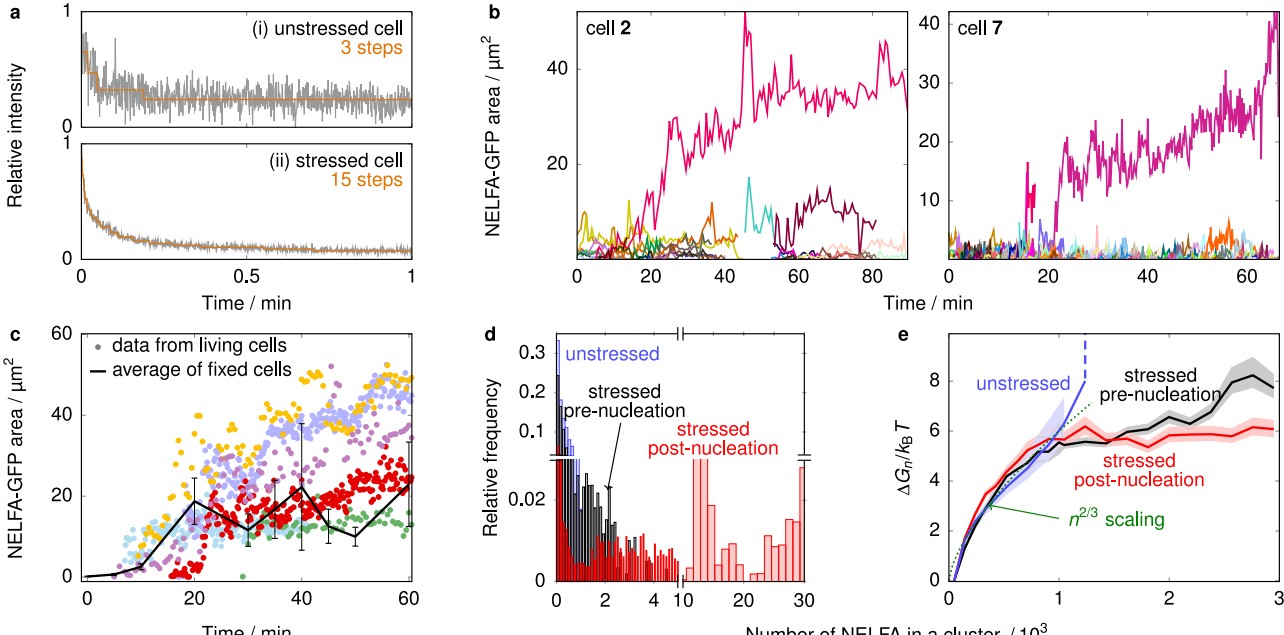

**Fig. 2 | Quantification of dynamic transient clusters. a** For conversion of areas of GFP regions into numbers of NELFA-GFP molecules, photobleaching steps were measured. Two example curves are shown for an unstressed cell (top) and a stressed cell (bottom), respectively. Steps (3 and 15, respectively) were determined by AutoStepfinder. **b** Time-resolved data for all clusters and condensates sizes of NELFA-GFP regions in two example living HeLa cells (see Supplementary Note 7 for the data of all cells). Individual clusters are shown in different colours. Most grow and shrink, and can reach sizes up to ~14 µm². Only once this size is exceeded do clusters grow steadily into a condensate (see main text for a quantification). **c** The black line gives sizes of clusters and condensates averaged over all 35 fixed HeLa cells, while circles are tracked sizes of NELFA-GFP regions in living HeLa cells. Each colour represents a distinct cluster that ultimately grows into a NELFA-GFP condensate in living-cell experiments. **d** The probability of a monomer being in a cluster of a certain size for stressed cells prior to nucleation, after nucleation started (post-nucleation) and for unstressed cells. **e** The free energy as a function of cluster size, computed from the probabilities in panel **d** (see main text for details). The centre line gives the mean over all the data values, and 99% confidence intervals shown are computed from 10,000 bootstrap samplings of data values. In **d** and **e**, data from seven stressed cells (1262 cluster sizes pre- and 3028 post-nucleation) and three unstressed cells (679 clusters) are shown. Numbers of NELFA are corrected for unlabelled protein.

between 25,000 and 125,000 NELFA proteins in the nucleus. This is within a small multiplicative factor of the number of NELFA in the nucleus, 155,688, determined by mass spectrometry[43], which therefore supports the quantitative nature of our analysis.

## NELF condensates are formed by non-classical nucleation

We showed above that condensate formation is a rare event and occurs only once a certain threshold size is reached [Fig. 2b, c]. To clarify the mechanism by which condensates form, we determined the lag time from the start of measurements to when fast growth to a large cluster size occurs [Supplementary Note 7]. These times are broadly distributed, which is a hallmark of a nucleation-and-growth mechanism[44]. However, the wide distribution of cluster sizes suggests that cluster growth is not controlled solely by competition between a favourable bulk term and a disfavourable interfacial term. Phenomenologically, the varied shapes of clusters [Fig. 1g and Supplementary Movie 4] may also indicate that the formation of an interface between the dense and dilute phases does not result in a large disfavourable effective free energy. A relatively low effective cost of interface formation might result from the thermodynamics of protein solutions, but may also arise from a more complex mechanism, for example, from mechanical stresses within the cell.

To probe the nature of the nucleation process further, we divided cluster trajectories into those that occurred prior to the fast condensate growth (pre-nucleation) and those that occurred afterwards (post-nucleation) [Supplementary Note 7]. This threshold is defined separately for each nucleus, accounting for the heterogeneity of the cells. For each scenario, we computed the probability $p_n$ that a monomer is in a cluster of a particular size $n$ [Fig. 2d], and, in turn, a Landau free-energy difference between clusters of a certain size

relative to monomers as $\Delta G_n = -k_B T \ln(p_n/np_1)$. These cluster sizes account for both labelled and unlabelled proteins. The resulting free-energy landscape [Fig. 2e] does initially increase with a typical surface scaling (~$n^{2/3}$), but then plateaus, suggesting the system may have a broad range of effective interaction strengths[45] and that the driving force for cluster growth is governed by a more complex mechanism than in classical nucleation theory, in which the free-energy decreases with a typical volume scaling (~$-n$). This plateau may, in part, also reflect the heterogeneity of the cellular environment; if the free energy increases for some (pre-critical) cells but decreases for post-critical ones, the overall average may appear flat, highlighting the importance of tracking and analysing individual cells. Finally, for unstressed cells, where no condensate formation has been observed, the free-energy barrier closely follows that of the stressed cells initially, but stops suddenly (i.e. it effectively diverges within the resolution of our experiments) and does not plateau. This suggests that, although the initial cluster growth is governed by a thermodynamic disfavourability of interface formation, subsequent cluster growth is blocked in unstressed cells. We cannot determine the mechanism for this blocking at this stage, but there can be many driving forces in the complex non-equilibrium environment of the nucleus, such as a loss of valency[46] or the blocking of the DNA sequences necessary for cluster formation[47] or other biochemical interactions[35]. Below, we show that p38 kinase plays a role in this mechanism.

Finally, we investigated further the mechanism of condensate growth post-nucleation. One possible mechanism by which large clusters could grow is Ostwald ripening, where medium-sized clusters gradually shrink as the largest one grows. This would result in a transient increase in the probability of both smaller and large clusters at the expense of medium-sized ones. However, we found that in our

case, cluster sizes were relatively evenly distributed across a wide range of cluster sizes, and the probability of a protein being in a small cluster did not increase as one large cluster grew (Supplementary Movie 6). Moreover, we determined the mean gradient of cluster size for every tracked cluster $j$ as $\nabla_{size}(j) = \frac{1}{n_j}\sum_{i=2}^{n_j}|\text{area}_i - \text{area}_{i-1}|$, where $n_j$ is the total number of steps over which cluster $j$ could be identified, and $\text{area}_i$ is the cluster's area at step $i$ of the trajectory. When we divided the gradient of cluster size by the area of the cluster [Supplementary Note 8 and Supplementary Fig. 9d], the resulting data fell within a narrow range even though the cluster size gradient itself was very much larger for large clusters than for smaller ones. This suggests that the addition of proteins does not occur stepwise; instead, larger clusters sweep up more of the smaller clusters via coalescence, as opposed to Ostwald ripening [Supplementary Note 9]. Similar behaviour has been observed for cortical condensates[35].

By combining fixed-cell data with living-cell imaging experiments, we can gain further insight into the real-time diffusion of all 4310 clusters investigated, for both stressed (3631) and unstressed (679) cells. Supplementary Fig. 9a shows the mean squared displacement of the centres of three example clusters over time from living-cell experiments. Although this graph already shows that the clusters do not only exhibit free Brownian motion, we initially calculated an effective diffusion coefficient $D$ from a fit to the Einstein relation, $\langle r^2 \rangle = 4Dt$ in the long-time limit. Both stressed and unstressed cells comprise clusters with a similarly broad distribution of diffusion coefficients [Supplementary Fig. 9b], while the background fluorescence can easily be distinguished from NELFA-GFP. To quantify the extent to which clusters diffuse freely, we also used a generalised diffusion equation $\langle r^2 \rangle = 4Dt^\alpha$ and determined the exponent $\alpha$. Supplementary Fig. 9a indicates that different diffusion mechanisms were in effect, ranging from free diffusion ($\alpha = 1$) to sub-diffusion ($\alpha < 1$) and directed diffusion ($\alpha > 1$). This range of behaviours is seen for both stressed and unstressed cells [Supplementary Fig. 9c] and underlines

the need to observe cluster formation in living cells, as such behaviour is not usually seen in model phase-separating systems.

## P38 kinase is required to form large clusters

The above physical analysis of cluster dynamics suggests that there might be a different regulation of small clusters compared to large clusters. To test this hypothesis, we investigated the effect of a p38 kinase inhibitor on the different clusters. P38 kinase has been shown to shuttle to the nucleus upon stress[38,48]. To ascertain whether the p38 kinase inhibitor interferes with NELF cluster formation, we incubated living cells with the p38 inhibitor for 1 h and then added $As_2O_3$ and observed the cells for 30 min under our HILO microscope. Figure 3a shows snapshots for an example cell [see Supplementary Movie 7 for this and two other cells], and Fig. 3c shows the distribution of the maximum cluster size from each tracked cluster. In the presence of the p38 kinase inhibitor, clusters larger than 600 NELFA are not formed, suggesting that p38 kinase is required for large cluster formation. In addition, this provides another way of distinguishing small and large clusters: we previously showed how cluster dynamics can be used as a criterion, but one could also use the susceptibility towards the p38 kinase inhibitor.

Finally, we tested what is required for the maintenance of large clusters. To this end, we first exposed cells to $As_2O_3$ for 30 min to form condensates and then added the p38 inhibitor in the presence of arsenic, i.e. the cells were still under stress. None of the condensates disappeared (four cells), but half the cells died. We next exposed cells to $As_2O_3$ for 30 min to form condensates, but now we removed the stress by exchanging the medium, and adding the p38 inhibitor; this led to several large clusters dissolving [Fig. 3b]. The cell-to-cell variation for this effect is high: 7 of the 13 investigated cells showed full or partial disappearance of large clusters, but some also stayed intact (see Supplementary Movie 8 for an example). By contrast, we never observed the dissolution of large clusters in the presence of arsenic.

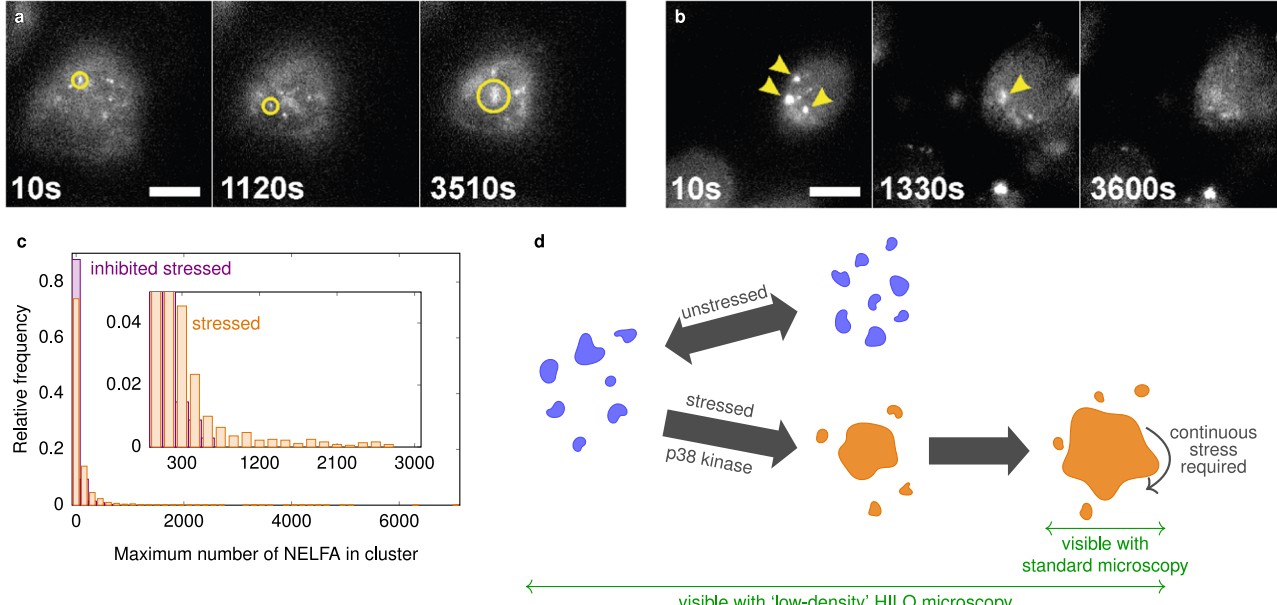

**Fig. 3 | P38 kinase is required to form large clusters. a** Cell nucleus of a cell that was treated with a p38 kinase inhibitor for 1 h prior to the addition of arsenic. Imaging with the HILO microscope began when $As_2O_3$ was added. Small clusters were assembled, but no large clusters were observed, even after 1 h of arsenic treatment. Three cells. **b** Cells were exposed to $As_2O_3$ for 30 min to form condensates (yellow arrows). The medium was exchanged (i.e., arsenic removed) to include p38 inhibitor at time 10 s, which caused several condensates (large clusters) to disassemble, while small clusters were still present. Images were corrected for photobleaching effects using the ImageJ plugin (Histogram Matching). A total of 13 cells were investigated; about half of them showed disassembly [see Supplementary Movie 8 for an example]. Disassembly of already formed large clusters was never observed in the presence of arsenic. All scale bars 10 μm. **c** Analysis of the maximum cluster sizes reached by each tracked cluster with and without pre-incubation with a p38 kinase inhibitor upon stress. The inhibitor interferes with the formation of large clusters, as no clusters larger than 600 NELFA were observed, providing another distinction between small and large clusters in living cells. Three cells; 344 clusters. **d** Schematic illustration of how small and large clusters respond to stress and how p38 kinase interferes with this process (see main text for details).

Altogether, our data show that p38 kinase is required to form large clusters under stress, and thus seems to be involved in unblocking the nucleation of NELF clusters in stressed cells. On the other hand, once large clusters have formed, they remain present for as long as the stress environment is kept, independent of the p38 kinase. Recent in vitro studies have shown that a reduced solubility of proteins increases the local density and favours phase separation[49], and that interactions driving cluster formation and larger-scale condensation can be separately tuned[29]. Our results indicate that p38 kinase is amongst the factors that enable such tuning in cells.

## Discussion

A combination of fluorescence imaging in living and fixed cells allowed us to quantify the dynamics of cluster and condensate formation for NELF in living cells. We obtained time-resolved NELF cluster sizes with almost single-protein resolution and could follow the clusters' dynamics in real time. This became possible through a unique combination of data from fixed and living cells. We showed that large stable clusters (condensates) only formed rapidly in the nucleus of stressed cells once a threshold of ~2000 NELFA proteins was reached. Strikingly, before condensate formation and even under conditions where no condensates form in the long-time limit, many smaller clusters of tens to hundreds of NELFA proteins were observed. Although this was predicted from the observation of heterogeneous distributions of clusters in subsaturated solutions in vitro[29], we found it rather unexpected that findings made in few-component in vitro systems and theoretically are directly applicable in the very complex environment of the cell.

Our physical analysis of all clusters in living cells shows that classical nucleation theory is insufficient to describe NELFA nucleation, which has recently also been found for condensate formation in vitro[29]. We obtained a relatively flat free-energy landscape following an initial surface-dominated barrier, with the probabilities of molecules being in clusters of many different sizes surprisingly similar. If we assume a similar free-energy landscape for stressed and unstressed cells, then our observation of a lack of a plateau in unstressed cells tells us that some process other than merely ordinary nucleation may be preventing further cluster growth. Blocking of nucleation, by whatever mechanism, phenomenologically corresponds to a large free-energy barrier. Effectively, this means that the free energy has increased very rapidly; indeed, so rapidly that no larger clusters are observed at all. This cannot be for technical reasons, because we detect these larger clusters easily in stressed cells.

Another possible explanation for the lack of a plateau in the free-energy landscape of unstressed cells may be that these two free-energy landscapes are simply fundamentally different and no effective comparison can be made. However, we consider this scenario to be less likely because the initial rise in the free energy is so similar.

The blocking of condensation could, to some extent, be caused by p38 kinase, as we have shown that it is involved in the release of the blocking of the formation of large NELF clusters in the nucleus. As a p38 inhibitor had no measurable effect on the small NELF cluster, the response to it provides another means of distinguishing between small and large clusters. Future experiments of the type we have presented will reveal if these transient small clusters are also regulated by chaperones, which have already been shown to modulate size distributions of self-associating proteins[50], even in an ATP-dependent manner[51].

In summary, we have shown that small, transient, dynamic clusters of NELFA appear before stable condensates form, even in unstressed cells. These small clusters differ from the large ones in their dynamics and in their response to the p38 kinase inhibitor. Therefore, the dynamics in cluster size as well as the response to the p38 kinase inhibitor present a means of unambiguously defining condensates, which is difficult for standard fluorescence threshold-based analysis. We expect that small dynamic clusters and the blocking of nucleation play an important role in cellular regulation and signalling. Such blocking may, for example, allow for a significant build-up of mass that can then result in rapid condensation when the cell requires it.

## Methods

### Cell culture and induction of NELFA-GFP expression

NELFA-GFP stable HeLa cells were grown in DMEM (Gibco 31053-028) to which 10% FBS (Gibco 10270-106), 100 units/mL penicillin (Sigma P4333), 100 µg mL$^{-1}$ streptomycin (Sigma P4333) and 2 mM L-glutamate (Sigma G7513) were added, at 37 °C and 5% CO$_2$[39]. All information on the source of the cell line used is given in ref. 39. Between $1 \times 10^4$ and $2 \times 10^4$ cells were seeded in Ibidi dishes (µ-dish 35 mm, high glass bottom dishes, Ibidi, 81158) with a refractive index of 1.52 to grow for 24 to 30 h before tetracycline induction. NELFA-GFP expression was induced by tetracycline (0.2, 0.4 and 1 µg mL$^{-1}$) for 4 to 6 h before living-cell imaging or fixation.

### HILO microscopy and living-cell imaging

Cells were imaged in the absence of As$_2$O$_3$ (unstressed) or while they were exposed to As$_2$O$_3$ (within 75 min, stressed) in live cell imaging solution (Invitrogen, A14291J) at 37 °C on a custom-built fluorescence microscope in HILO mode (objective: Nikon Apo TIRF 100×/1.49 oil) with an EMCCD camera (Andor iXon Ultra 897) at a laser (Coherent, 473 nm) excitation power density of 60 mW cm$^{-2}$ (for time-lapse living-cell imaging) or constant 240 mW cm$^{-2}$ (for measuring the photobleaching steps in fixed cells). The recorded area was 40.96 µm × 40.96 µm; see also Supplementary Note 3.

For our single-cell experiments, we selected HeLa cells in which NELFA-GFP fluorescent spots diffused in an apparently random manner. Controls have shown that the background fluorescent pattern was either static or moved in a directed manner. We immediately added 100 mM As$_2$O$_3$ (Sigma 202673-5G) at multiple positions of the Ibidi dish, which contained 2 mL live cell imaging solution, resulting in a final concentration of 100 µM of As$_2$O$_3$. The time-lapse was started when As$_2$O$_3$ was added, with a time interval of either 10 or 30 s between two frames. The camera exposure time of each frame was 70 ms.

For fixed-cell imaging for measuring photobleaching steps, HeLa cells were treated with 100 µM As$_2$O$_3$ for 60 min and were further fixed using Image-iT™ fixative solution (Invitrogen FB002) at room temperature for 15 min. Cells were then washed with DPBS (Gibco 14190144) five times, after which they were ready for imaging. The fixed cells for analysis were selected to have similar NELFA-GFP regions compared to the living cells at the respective time points.

Finally, we selected cells which had ideal expression conditions of NELFA-GFP for our fluorescence experiments, i.e. about one NELFA-GFP per four NELFA. We believe that this minimally perturbs the wild-type system, which is supported by our viability assays. In addition, every single cell from our living-cell experiments was observed for about 60 min. Therefore, the full set of results (dynamic size, diffusion coefficients) was obtained for every single cell, without averaging. We do not claim that every single cell shows exactly this dynamic behaviour, but many cells do, and in total, we have investigated more than 60 cells.

To exclude imaging artefacts, we measured a control system (2NT-DDX4-GFP)[52] with our setup and found spherical condensates [Supplementary Fig. 3c] consistent with previous descriptions[52].

### Analysis of living-cell imaging movies

Recorded movies were first processed using the Weka segmentation plugin in Fiji[53] to extract NELFA-GFP regions from the manually assigned cellular and non-cellular background and were further processed using the Mosaic plugin[54,55] in Fiji to track NELFA-GFP regions [Supplementary Note 4]. Further data were evaluated and plotted using the ImageJ macro (open source) and MATLAB (MathWorks) using custom code. Some further analysis was performed with

standard Unix command-line tools, including GNU Awk 5.0.1, Gnuplot 5.2 patchlevel 8, Perl 5.30.0, Python 3.8.10 (with SciPy 1.5.2, matplotlib 3.3.2, NumPy 1.17.4, KDEpy 1.1.0), GNU bash 5.0.17(1), GNU sed 4.7, GNU grep 3.4, GNU findutils 4.7.0 and GNU coreutils 8.30; all scripts are provided as part of the supporting code[56].

### Analysis of fixed-cell imaging movies
Recorded movies were first analysed using Fiji, ImageJ macro and MATLAB custom code to extract locations and bleaching curves in dilute and dense phases with background correction [Supplementary Note 6]. Photobleaching steps were measured using AutoStepfinder[57].

### Immunofluorescence microscopy
NELFA-GFP HeLa cells seeded on Ibidi chambers were treated with tetracycline for 4 h to induce the expression of NELFA-GFP. Some cells were further exposed to 100 μM $As_2O_3$, as described previously. Cells were washed with PBS and fixed in 4% paraformaldehyde in PBS for 10 min at room temperature, followed by permeabilisation with 0.3% Triton X-100 in PBS for 10 min at room temperature. Cells were washed with PBS and incubated with DAPI (Sigma, #D9542; 1:1000 from 0.5 mg mL$^{-1}$ stock) for 5 min at room temperature. Thereafter, cells were washed with PBS and stored at 4 °C before imaging. Fluorescence images were generated using a Zeiss LSM800 microscope equipped with a 63×, 1.4 NA oil objective and an Airyscan detector and processed with Zen blue software and ImageJ/Fiji. Cells were imaged as z-stack with 130-nm sections with a lateral resolution of 120 nm.

### Reporting summary
Further information on research design is available in the Nature Portfolio Reporting Summary linked to this article.

## Data availability
All data generated or analysed during this study are included in this published article and its supplementary files. Supporting movies and raw data are included on Zenodo at https://doi.org/10.5281/zenodo.6946007[56].

## Code availability
All custom code used in this study is included on Zenodo at https://doi.org/10.5281/zenodo.6946007[56].

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

## Acknowledgements
We thank Ibrahim Cissé, Rosana Collepardo-Guevara, Stephan Grill and Rohit Pappu for helpful discussions, and Adam Klosin (Hyman lab) for supplying us with the plasmid of DDX4. This work was supported by the European Research Council (grant agreement No. 681891, T.H.) and the Deutsche Forschungsgemeinschaft (DFG) under Germany's Excellence Strategy (CIBSS EXC-2189 Project ID 390939984, R.G., R.S. and T.H.) and the SFB1381 programme (Project ID 403222702, T.H.).

## Author contributions
A.R., R.S. and T.H. designed the research; C.L., J.K., F.A.-G. and S.U. performed the measurements; C.L., S.W., A.R., R.S. and T.H. analysed the data after consultation with R.G. and A.A.; C.L., A.R., R.S. and T. H. wrote the manuscript. All authors discussed the results and commented on the manuscript.

## Funding

## Competing interests
The authors declare no competing interests.
