## [Peer Review file · Nature Communications]

REVIEWER COMMENTS

Reviewer #1 (Remarks to the Author):

In this manuscript by Lan et al. the authors use a new high-resolution, low-light microscopy technique to follow phase-separating NELF molecules before phase separation, during the growth phase and after phase separation. They show that NELF forms clusters that can grow or shrink, and that they only grow decisively after reaching a critical size. They also show that condensates grow through coalescence more so than Ostwald ripening. Finally, they look at the effect of an inhibitor of p38 kinase and can show that p38 kinase activity is not required for the formation of small clusters but of the formation of larger condensates.

These types of pre-percolation clusters have previously been shown in vitro for simple systems but never in cells. The current work beautifully enables quantification of how they grow and shrink, and eventually grow into condensates when they reach a critical size. The findings in the current manuscript support the recent findings made experimentally in vitro and theoretically and via computation and show that they are directly applicable also in cells. This provides a framework through which to conceptualize cellular phase separation, and I predict that it will help break the phase separation field through its largely phenomenological stage. I thus find this an important manuscript and support publication of this work.

The authors have addressed my comments to my satisfaction.

Reviewer #2 (Remarks to the Author):

The authors provide quantitative analysis of stress-induced NELFA cluster formation in living and fixed cells. Importantly, they for the first time give comprehensive analysis of cluster behavior in conditions where no condensates are able to form and in conditions where condensation occurs. This is possible through fine-tuning of protein expression and conditional addition of arsenic stress.

Clusters are quantified both in terms of diffusion coefficient and cluster size as function of time. This is achieved by tracking individual clusters over time, and calibrating their size using fixed cell data.

Moreover, the authors present data regarding p38 kinase, which is involved in unlocking condensate formation but not necessarily condensate maintenance.

This is a very important contribution to the field of transcription factor condensates. It is well written, technically sound (but see below for tracking) and all methods are well described. I support its publication. However, some points have not yet been commented by previous reviewers and need to be addressed prior to publication.

Major point:

- While the analysis of cluster identification and size determination is very rigorous, the tracking of clusters over time does not seem to work well for all clusters. It appears quite some clusters which do not visually belong together are connected. This is not surprising given the long intervals of 10s between images, and the dynamic behavior of clusters. In particular, the large steps seen in many movies are at least questionable.

The authors need to comment on the accuracy of their tracking algorithm. How often do they estimate that wrong clusters are connected? How do false connections affect the results?

Minor points:

- supplementary movie 1 shows a red color, in addition to blue and green, but it is not stated where this comes from

- supplementary movie 3 does not seem to be the same cell as the one presented in figure 1f

- figure 2 d/e: scaling should be 10^3 instead of 10^{-3} ?

- figure 2e: how was bootstrapping performed?

- figure 2 d/3 and figure 3c: it is unclear whether the given numbers are already corrected for unlabeled protein or not

- line 303: increase should be decrease?

- the statement about p38 is sometimes too strong. While it certainly has an effect, it need not be the only factor involved. Thus, the authors should rephrase line 380: ... p38 kinase is amongst the factors that enable... or similar. Line 422: ... could to some extent be caused by p38 kinase... or similar

- figure S10: cell 9 does not seem to have a clear growing condensate? But a time of 7 min is given for start of growth.

- figure S12c: The numbers of 679 control and 184 unstressed tracks might be interchanged, please check.

Reviewer #3 (Remarks to the Author):

My review was requested specifically in regards to the image analysis work presented here; I therefore do not comment on anything else in the paper.

Unfortunately, I cannot fully reproduce the authors tracking solution, since neither raw data nor the Weka model were provided. However, I believe based on the provided example movies that their tracking parameters are likely not clean enough to allow tracking of small, fast moving particles over time (the big ones seem to do more or less ok), as my attachments show. This makes all figures based on tracking information suspect. They might do better using something like Cellpose plus Trackmate, and/or maybe something like TracX after Cellpose segmentation. LAP tracking is designed for this kind of problem, and is available in multiple tools (ImageJ, CellProfiler, etc). Most of their biological questions, though, seem to me to not required tracking - only supplemental figures 11-13 require it, and I'm not sure that saying "we see all kinds of different diffusion coefficients" adds that much biologically, and particle size histograms at various time points I think in general get at most of the other biology they care about.

It is also not clear to me how their photobleaching background subtraction calculations work - it is not described how the background fluorescence is calculated in the dense phase (presumably the "background fluorescence" calculation is unreliable in the middle of a condensate where there is no true "background" present), and I'm not sure, if the intensities represented in their equation truly represent MEAN and not TOTAL intensities, why the background needs to be divided by 9/16. The next paragraph also states that "unfiltered" data goes into the step finding algorithm, so what is the purpose of this step?

More explanation is needed for supp figure 12 c, it's not described in any detail where this number comes from.

Reviewer #4 (Remarks to the Author):

The authors have tried to address my concerns. I must say that I am still largely unconvinced by many of the claims in the paper. I do not wish to bring up new problems (as I do not seem to have noticed this particular issue in the earlier version), but the photobleaching "counting" data are particularly unconvincing, and the notion that there are "15 steps" in the data displayed in Fig 2A (bottom) is completely beyond belief, especially given that the step sizes shown in the step finding algorithm are allowed to vary.

Quantitative real-time in-cell imaging reveals heterogeneous clusters of proteins prior to condensation

NCOMMS-23-04343-T

Point-by-point response

In our responses below, we have used the same figure numbering scheme as in our initial submission, but we also give the new figure number in square brackets. When revising the manuscript, we were requested to comply with the journal's formatting instructions and we have therefore renumbered the figures in the supplementary information document to start counting from 1 rather than continuing the count from the main text. In other words, what was previously Fig. S4 is now Fig. S1, and so forth.

Referee 1's comments

In this manuscript by Lan et al. the authors use a new high-resolution, low-light microscopy technique to follow phase-separating NELF molecules before phase separation, during the growth phase and after phase separation. They show that NELF forms clusters that can grow or shrink, and that they only grow decisively after reaching a critical size. They also show that condensates grow through coalescence more so than Ostwald ripening. Finally, they look at the effect of an inhibitor of p38 kinase and can show that p38 kinase activity is not required for the formation of small clusters but of the formation of larger condensates.

These types of pre-percolation clusters have previously been shown in vitro for simple systems but never in cells. The current work beautifully enables quantification of how they grow and shrink, and eventually grow into condensates when they reach a critical size. The findings in the current manuscript support the recent findings made experimentally in vitro and theoretically and via computation and show that they are directly applicable also in cells. This provides a framework through which to conceptualize cellular phase separation, and I predict that it will help break the phase separation field through its largely phenomenological stage. I thus find this an important manuscript and support publication of this work. The authors have addressed my comments to my satisfaction.

We thank the referee for a great summary of our work and for supporting publication of the manuscript as it stands.

Referee 2's comments

The authors provide quantitative analysis of stress-induced NELFA cluster formation in living and fixed cells. Importantly, they for the first time give comprehensive analysis of cluster behavior in conditions where no condensates are able to form and in conditions where condensation occurs. This is possible through fine-tuning of protein expression and conditional addition of arsenic stress. Clusters are quantified both in terms of diffusion coefficient and cluster size as function of time. This is achieved by tracking individual clusters over time, and calibrating their size using fixed cell data.

Moreover, the authors present data regarding p38 kinase, which is involved in unlocking condensate formation but not necessarily condensate maintenance.

This is a very important contribution to the field of transcription factor condensates. It is well written, technically sound (but see below for tracking) and all methods are well described. I support its publication. However, some points have not yet been commented by previous reviewers and need to be addressed prior to publication.

We thank the referee for their clear articulation of the main message of our work and for their support in its publication. We address the specific points they have raised below.

Mayor point:

- While the analysis of cluster identification and size determination is very rigorous, the tracking of clusters over time does not seem to work well for all clusters. It appears quite some clusters which do not visually belong together are connected. This is not surprising given the long intervals of 10s between images, and the dynamic behavior of clusters. In particular, the large steps seen in many movies are at least questionable. The authors need to comment on the accuracy of their tracking algorithm. How often do they estimate that wrong clusters are connected? How do false connections affect the results?

We agree with the referee (and with referee 3) that tracking is difficult for our system. We have therefore performed additional tracking analyses. Specifically, we have now confirmed our tracking results by manually checking more than 1000 clusters and their tracks and noted which clusters would be assigned differently from the algorithm. Three cases were distinguished: (1) A track was interrupted ($< 3\%$); (2) A track of one cluster continued as another cluster ($< 6\%$); (3) Two clusters were counted as one or one cluster was counted as two ($< 10\%$). These errors are relatively small and we can therefore have confidence in the qualitative conclusions concerning diffusion coefficients. We have added a brief discussion of this tracking analysis to Supplementary Note 8, where we now explicitly explain that although there is some scope for misidentification, this cannot on its own explain the large spread in diffusion coefficients.

For the cluster sizes and in turn the free-energy profiles, where only size counts rather than tracking are important, the effect is even more minor, because (i) the numbers of clusters misidentified that ended up being smaller and larger than the correct sizes were roughly equal, and (ii) the clusters affected were predominantly small: since there are very many small clusters to begin with, small differences in their counts are therefore the least important.

Minor points:

- supplementary movie 1 shows a red color, in addition to blue and green, but it is not stated where this comes from

We thank the referee for pointing out this omission. We have now added the information that the red colour in movie 1 is a stain for the nuclear lamina with AF647.

- supplementary movie 3 does not seem to be the same cell as the one presented in figure 1f

We indeed had the wrong movie uploaded as movie 3; we thank the referee for spotting this and have now uploaded the correct movie.

- figure 2 d/e: scaling should be 10^3 instead of 10^{-3} ?

The ' $\times 10^{-3}$ ' was part of the axis label, using the quantity calculus approach recommended by IUPAC [see for example p. 3 of the Green Book]. If we label the number of NELFA in a cluster as N , then the x axis gives $N \times 10^{-3}$. When the number on the x axis is, say, 1, $10^{-3}N = 1$, or $N = 10^3$. This should in principle mean that all quantities are unambiguous, but clearly it has caused some confusion in this instance. We have now changed the axis label in Fig. 2d,e to ' $/10^3$ ', which is still consistent with quantity calculus, but can also be read as a scaling factor, thereby minimising the possibility of being misunderstood.

- figure 2e: how was bootstrapping performed?

In order to estimate error bars by bootstrapping, for each case considered, we started with the cluster size data obtained from trajectory analyses; the number of recorded data points was 4359 (unstressed), 8853 (stressed pre-nucleation) and 21737 (stressed post-nucleation). We created 10000 bootstrap resamples, each comprising the same total number of data points as the original data, by randomly drawing samples (with replacement). We computed cluster-size histograms for each of the 10000 realisations to give a bootstrap distribution of histogram data, and finally we collated all binned data for the same bin, sorted it, and extracted the 50th and the 9950th data point (out of 10000) to give the 99% bootstrap confidence interval. The script used to generate these intervals is included in our supporting data archive. We now explain this procedure in full in the readme file associated with the supporting data archive.

- figure 2 d/3 and figure 3c: it is unclear whether the given numbers are already corrected for unlabeled protein or not

We appreciate that our distinction between 'NELFA-GFP' (the labelled protein) and 'NELFA' (corrected for the unlabelled fraction) was, in retrospect, less clear than it could have been. We have now clarified in the manuscript (in the caption of Fig. 2) that protein *numbers* (as in these figures) are corrected for unlabelled protein. The *areas* correspond to the raw data (i.e. NELFA-GFP only).

- line 303: increase should be decrease?

If Ostwald ripening were occurring, then we would expect the broad distribution of clusters to start splitting into large and small clusters, as proteins dissolve from 'medium-sized' clusters to provide the building blocks of large clusters. The medium-sized clusters thus shrink, and at least initially the distribution would thus be expected to become more bimodal and the probability of small clusters would increase. We have now added a more explicit statement why the probability of small clusters is expected to increase by adding the following sentence just before the sentence the referee has highlighted: 'This would result in a transient increase in the probability of both smaller and large clusters at the expense of medium-sized ones.'

- the statement about p38 is sometimes too strong. While it certainly has an effect, it need not be the only factor involved. Thus, the authors should rephrase line 380: ... p38 kinase is amongst the factors that enable... or similar. Line 422: ... could to some extent be caused by p38 kinase... or similar

We have followed the referee's guidance and used the exact phrasing they have suggested. Namely, we have replaced the sentence 'It may be that it is p38 kinase that enables such a tuning in cells' with 'Our results indicate that p38 kinase is amongst the factors that enable such a tuning in cells', and we have replaced the clause 'The blocking of condensation could be caused by P38 kinase' with 'The blocking of condensation could to some extent be caused by p38 kinase'.

- figure S10: cell 9 does not seem to have a clear growing condensate? But a time of 7 min is given for start of growth.

Indeed, this growth phase is rather less clear than the rest. We took the cluster shown in a blue hue (that grows at around the 7 min mark) as a condensate because it remains stable in size throughout the 10 min to 40 min range, i.e. it does not fit into the category of the small dynamic clusters. On the other hand, as the referee points out, larger-scale growth is also not clear. We have added this caveat to the figure caption.

- figure S12c: The numbers of 679 control and 184 unstressed tracks might be interchanged, please check.

We thank the referee for pointing this out. We had indeed interchanged the numbers, and have now fixed this.

Referee 3's comments

My review was requested specifically in regards to the image analysis work presented here; I therefore do not comment on anything else in the paper.

Unfortunately, I cannot fully reproduce the authors tracking solution, since neither raw data nor the Weka model were provided. However, I believe based on the provided example movies that their tracking parameters are likely not clean enough to allow tracking of small, fast moving particles over time (the big ones seem to do more or less ok), as my attachments show. This makes all figures based on tracking information suspect. They might do better using something like Cellpose plus Trackmate, and/or maybe something like TracX after Cellpose segmentation. LAP tracking is designed for this kind of problem, and is available in multiple tools (ImageJ, CellProfiler, etc). Most of their biological questions, though, seem to me to not required tracking - only supplemental figures 11-13 require it, and I'm not sure that saying "we see all kinds of different diffusion coefficients" adds that much biologically, and particle size histograms at various time points I think in general get at most of the other biology they care about.

We thank the referee for their helpful suggestions to optimise tracking. We have now tried Cellpose and LAP tracking. Cellpose and LAP are designed for live cell tracking of cell lineages having a similar shape and size. We found that both methods cannot provide tracking as reliable as the Weka segmentation since our particles constantly reshape and resize over time. Having talking to other tracking experts, we believe the main difficulty in using various ready-made tools is that all these tracking algorithms are not intended to track particles that change their size and shape. We have therefore decided to confirm our tracking results by manually checking more than 1000 clusters and their tracks and noting which clusters would be assigned differently from the algorithm. Three cases were distinguished: (1) A track was interrupted (< 3 %); (2) A track of one cluster continued as another cluster (< 6 %); (3) Two clusters were counted as one or one cluster was counted as two (< 10 %).

The errors are relatively low and therefore they cannot on their own result in the large spread in diffusion coefficients that we observe. As we believe that, although this piece of information is not, as the referee notes, central to the argument, it is nevertheless a useful remark to make, we would like to retain the comment on the large spread in diffusion coefficients. We have therefore added a brief discussion of this tracking analysis to Supplementary Note 8, where we now explicitly explain that although there is some scope for misidentification, this cannot on its own explain the large spread in diffusion coefficients.

As the referee expected, for the cluster sizes and in turn the free-energy profiles, where only size counts rather than tracking are important, the effect is even more minor, because (i) the numbers of clusters misidentified that ended up being smaller and larger than the correct sizes were roughly equal, and (ii) the clusters affected were predominantly small: since there are very many small clusters to begin with, small differences in their counts are therefore the least important.

The raw data are contained in the supplementary movies, but we appreciate that it may not be entirely straightforward to extract the data from the movies themselves. We have therefore uploaded the TIFF stacks for every movie on our

server at <https://nas.physchem.uni-freiburg.de:5679/sharing/vv9BPhQFG> for the referees' personal use, and we will also upload these to Zenodo once the manuscript is accepted for publication. The Weka segmentation is a free Fiji plugin available at <https://imagej.net/plugins/tws/how-to-install-new-classifiers>.

It is also not clear to me how their photobleaching background subtraction calculations work - it is not described how the background fluorescence is calculated in the dense phase (presumably the "background fluorescence" calculation is unreliable in the middle of a condensate where there is no true "background" present), and I'm not sure, if the intensities represented in their equation truly represent MEAN and not TOTAL intensities, why the background needs to be divided by 9/16. The next paragraph also states that "unfiltered" data goes into the step finding algorithm, so what is the purpose of this step?

Yes, the intensities in the equation for the background correction in Note 6 are indeed mostly total intensities and not mean intensities; our apologies. What we meant to say is: We first take the (total) intensity of a 3×3 pixel area, then we subtract the (total) intensity of a rim of 16 pixels around this 3×3 pixel region. To normalise this rim, we have to multiply by 9/16. Finally, we subtract the (mean) intensity from 10 randomly selected 3×3 pixel areas outside the cell ($\langle i_{\text{non-cell}} \rangle$). We have clarified this in the revised version. Indeed, for the step counting, this background subtraction does not play a role. We have now also performed a brightness analysis of cluster sizes. In particular, we determined the mean intensity for single photobleaching steps and then determined the total brightness of every cluster and obtained the number of GFP by their ratio. We have added these data to Fig. S9c [now Fig. S6c]. The error in the absolute number of NELFA-GFP is less than 30 %, giving us confidence that the claims of our manuscript hold.

More explanation is needed for supp figure 12 c, it's not described in any detail where this number comes from.

In Fig. S12c [now Fig. S9c], the 'unstressed' data are the same as in Fig. S12b [now Fig. S9b]. We then added in green the data from three control cells (i.e. with no NELF-GFP) obtained by the tracking of background fluorescence, i.e. background fluorescent spots. The reason for these experiments is to find an independent criterion to distinguish NELF-GFP from other fluorescent spots. We have now further clarified this in the figure caption.

Referee 4's comments

The authors have tried to address my concerns. I must say that I am still largely unconvinced by many of the claims in the paper. I do not wish to bring up new problems (as I do not seem to have noticed this particular issue in the earlier version), but the photobleaching "counting" data are particularly unconvincing, and the notion that there are "15 steps" in the data displayed in Fig 2A (bottom) is completely beyond belief, especially given that the step sizes shown in the step finding algorithm are allowed to vary.

We hope that the more thorough tracking analysis, as suggested by referees 2 and 3, now substantiates our claims further. In addition, in response to the referee's specific point, we have further benchmarked the counting of numbers of GFP in a cluster comparing the step-counting method with an intensity-based method in our revised manuscript.

First, we should like to emphasise that AutoStepfinder, which we have used to count the photobleaching steps, was thoroughly tested and the detection limits are well below the signal-to-noise ratio of our data; see e.g. Fig. 5a of Loeff et al. [Patterns 2, 100256 (2021)].

As a second method, in the revised manuscript, we determined the number of GFP in a cluster not from the number of bleaching steps, but from the intensity of a cluster. Specifically, we first determined the mean intensity for single photobleaching steps, which can very readily be quantified and detected (see e.g. Fig. 2a, top). Then we determined the total brightness of every cluster and obtained the number of GFP by computing their ratio. We have added these data from this additional method to Fig. S9c [now Fig. S6c], which shows that the error in the absolute number of NELFA-GFP is less than 30 %. Thanks to the referee's comment, we have thus been able to add an estimate of the error bar in our quantification of cluster sizes, and as a consequence we can have even more confidence in the claims made in our manuscript.

REVIEWER COMMENTS

Reviewer #2 (Remarks to the Author):

The authors have addressed and resolved all my concerns. I support publication of this important work clarifying aspects of condensate formation.

Reviewer #3 (Remarks to the Author):

While I appreciate the authors' hard work in manually reviewing their tracks, I still find aspects of their tracking approach problematic. If these remaining issues are addressed, my limited critique of the image analysis aspect is satisfied.

- I do not think trying to do fluorescence tracking of "background fluorescence spots" in cells not expressing fluorescent protein serves in any way as a useful control. Since such areas are dim it would presumably be highly prone to error, and since it's not a specifically tagged object, you have no idea what it is or how it would be expected to move.

- I apologize for being unclear, I was not saying to the authors that I didn't know how to install Weka, I'm saying it's impossible to fully reproduce their approach unless they save and provide their classifier(<https://imagej.net/plugins/tws/#save-classifier>) and/or annotations (<https://imagej.net/plugins/tws/#save-data>) for others to use. If I have overlooked their .model file or .ARFF files, I apologize, and would be glad to have a pointer to it.

With regards to the photobleaching, I still don't understand in what experiment/figure the "take the 3x3 grid and background subtract" approach is used, since the authors clarify that's not used in stepfinding and that's the only way of quantifying photobleaching that as far as I can tell persists from the previous manuscript.

Quantitative real-time in-cell imaging reveals heterogeneous clusters of proteins prior to condensation

NCOMMS-23-04343-A

Point-by-point response

Referee 2's comments

The authors have addressed and resolved all my concerns. I support publication of this important work clarifying aspects of condensate formation.

We are very pleased to hear that we have been able to address the referee's concerns to their satisfaction.

Referee 3's comments

While I appreciate the authors' hard work in manually reviewing their tracks, I still find aspects of their tracking approach problematic. If these remaining issues are addressed, my limited critique of the image analysis aspect is satisfied.

We thank the referee for their careful further reading of our manuscript, and we apologise that our previous revision left three points open, which we are happy to address in the following.

I do not think trying to do fluorescence tracking of "background fluorescence spots" in cells not expressing fluorescent protein serves in any way as a useful control. Since such areas are dim it would presumably be highly prone to error, and since it's not a specifically tagged object, you have no idea what it is or how it would be expected to move.

The referee is entirely correct to point out that we do not know what the 'background' intensity corresponds to. Indeed there is also likely to be a significant error associated with the determination of their size, and only relatively short trajectories can be obtained. The original idea was to have an additional means to estimate the effect of the background on our data analysis, especially the diffusion analysis. We thought it may be useful to show that, even in this regime where ballistic motion may play a significant role, the effective 'diffusion' coefficients of the background are well separated from the clusters. However, we agree with the referee that this does not really provide any useful information beyond saying that we can readily distinguish between the bright fluorescent spots and the background.

As advised, we have removed what was Fig. S9c (and references to it). Figs S9d and S9e are now labelled S9c and S9d, respectively.

I apologize for being unclear, I was not saying to the authors that I didn't know how to install Weka, I'm saying it's impossible to fully reproduce their approach unless they save and provide their classifier(<https://imagej.net/plugins/tws/#save-classifier>) and/or annotations (<https://imagej.net/plugins/tws/#save-data>) for others to use. If I have overlooked their .model file or .ARFF files, I apologize, and would be glad to have a pointer to it.

We apologise for the misunderstanding. We have now included a full description of the Weka analysis pipeline in our 'Notes on the supporting code and data' (see third paragraph of the Analysis Pipeline in the Supporting-code-and-data-README.pdf). In addition, we provide for every analysed movie the Classified image, the Classifier model, the ARFF file and the Weka input TIFF. Everything can be found at the following link:

<https://cloud.physchem.uni-freiburg.de/s/YJQdXFdw9fQTafS> (password: Weka27_06_23). Once the manuscript is accepted for publication, we will upload a zip file of this folder onto Zenodo.org.

With regards to the photobleaching, I still don't understand in what experiment/figure the "take the 3x3 grid and background subtract" approach is used, since the authors clarify that's not used in stepfinding and that's the only way of quantifying photobleaching that as far as I can tell persists from the previous manuscript.

Our apologies; this '3 × 3 grid and background subtract' is indeed a leftover from a previous version of this manuscript, where we tried to directly account for the background of 3 × 3 grids. This is no longer necessary for the current photobleaching-based analysis. We have now removed this part of the supplementary note.

REVIEWERS' COMMENTS

Reviewer #3 (Remarks to the Author):

I thank the reviewers for addressing my concerns.